# Community perception of school-based mass drug administration program for soil-transmitted helminths and Schistosomiasis in Ogun State, Nigeria

Folahanmi T. Akinsolu[1,2]*, Olunike R. Abodunrin[1,3], Mobolaji T. Olagunju[4], Ifeoluwa E. Adewole[1‡], Nurudeen O. Rahman[5], Anita M. Dabar[1‡], Diana W. Njuguna[6], Islamiat Y. Soneye[7‡], Abideen O. Salako[1,2]*, Oliver C. Ezechi[1,2]*, Orsolya E. Varga[8], Olaoluwa P. Akinwale[1,2]

1 Lead City University, Ibadan, Nigeria, 2 Nigerian Institute of Medical Research, Lagos, Nigeria, 3 Lagos State Health Management Agency, Lagos, Nigeria, 4 Nanjing Medical University, Nanjing, China, 5 Swiss Tropical and Public Health Institute, Allschwil, Switzerland, 6 Dedan Kimathi University of Technology, Nyeri, Kenya, 7 Ogun State Ministry of Health, Abeokuta, Nigeria, 8 University of Debrecen, Debrecen, Hungary

☯ These authors contributed equally to this work.
‡ IEA, AMD, IYS and CD also contributed equally to this work.
* Folahanmi.tomiwa@gmail.com (FTA); salako.abideennaheem@gmail.com (AOS); oezechi@yahoo.co.uk (OCE)

**Data Availability Statement:** All relevant data are within the manuscript and its supporting information files.

## Abstract

### Background

Neglected tropical diseases, such as soil-transmitted helminths and Schistosomiasis, are prevalent in sub-Saharan Africa, particularly Ogun State, Nigeria. School-based mass drug administration program is the primary control intervention, but the coverage and uptake of this intervention have been inadequate. This study aimed to investigate community perceptions of school-based mass drug administration programs for these infections in Ogun State, Nigeria, and identify the barriers to their uptake and coverage.

### Methodology/Principal findings

The study used a qualitative research approach involving focus group discussions and in-depth interviews with community members and stakeholders engaged in neglected tropical disease control programs in Ogun State, Nigeria. A semi-structured questionnaire guided the exploration of ideas, and the data were analyzed using the QRS Nvivo 12 software package. The study found several barriers, such as the influence of parents, lack of sufficient knowledge, and side effects. The study recommended strategies such as improving community sensitization and engagement, drug distribution and performance, and enhancing partner collaboration and coordination to improve the school-based mass drug administration programs.

**Funding:** This work received financial support from the United States Agency for International Development (USAID) through its Neglected Tropical Diseases Program of through their support of the Coalition for Operational Research on Neglected Tropical Diseases (COR-NTD) grant to FTA. COR-NTD is funded at The Task Force primarily by the Bill & Melinda Gates Foundation and USAID. The funders had no role in study design, data collection and analysis, decision to publish, or preparation of the manuscript.

**Competing interests:** The authors have declared that no competing interests exist.

## Conclusions/Significance

The study revealed correct perceptions of transmission but some misconceptions about disease causation, transmission, and drug safety. Participants expressed a desire for better sensitization campaigns and more assurances of their safety. The study recommends strengthening health education messages and increasing the visibility of on-site medical personnel. The findings have implications for improving the performance of these programs and reducing the burden of intestinal parasitic infections in the community. The study highlights the need for community engagement and education, health system support, and partner collaboration to successfully implement mass drug administration programs.

### Author summary

This study explored the barriers to the uptake and coverage of control interventions for soil-transmitted helminths and Schistosomiasis in Nigeria. The study used key informant interviews and focus group discussions were conducted with stakeholders involved in Neglected Tropical Disease school-based control programs, including community members, teachers, parents, and school-aged children. The study found that the main barriers to the uptake and coverage of control interventions for soil-transmitted helminths and schistosomiasis were poor drug acceptability, limited accessibility to drugs, and inadequate knowledge about the diseases and the control interventions. Additionally, the study found that the implementation of Neglected Tropical Disease control programs was inconsistent due to the inadequacy of available support from partners. Overall, our study provides important insights into the barriers and facilitators to Neglected Tropical Disease school-based control programs and highlights the need for improved drug acceptability, accessibility, and knowledge about the diseases and control interventions. Our findings can inform the development of effective interventions to improve the uptake and coverage of control interventions.

## Introduction

Soil-transmitted helminths (STH) and Schistosomiasis (SCH) are among 20 neglected tropical diseases (NTDs) identified by WHO and are responsible for approximately 150,000 deaths a year [1–3]. Africa accounts for 40% of the global burden of NTDs [4]. NTDs are major public health problems in several developing countries, including Nigeria which has the highest prevalence in sub-Saharan Africa [2,5]. In Nigeria, STH and SCH infections harm school-aged children's health and fitness (SAC) [6].

   Over the last three decades, considerable attention and funding have been given to NTDs, with most recent efforts being targeted towards the attainment of commitments detailed in the WHO, Ending the neglect to attain the Sustainable Development Goal that pursues the control and elimination of selected NTDs by 2030 [7,8]. STH and SCH are primarily controlled through mass drug administration (MDA) programs delivered to SAC via school-based delivery platforms to reduce morbidity's occurrence, severity, and long-term consequences [9]. While MDA is the primary strategy for controlling and eliminating STH and SCH, the treatment has been inconsistent. In addition, there are concerns regarding the ability of these intervention achievements to be maintained in the face of programmatic, social, and political

changes [2,10]. With the approaching year 2030, it is imperative to assess both the accomplishments and obstacles encountered in the implementation of MDA to ensure that the goal of equitable achievement of the 'end game' is attained in varied and rapidly evolving contexts.

A better understanding of the facilitators and barriers of integration within existing integrated programs can enhance public health integration efforts in Nigeria. The integration of NTD control programs has been identified as a promising approach toward achieving the London Declaration goals and has been recommended by the WHO for NTD-endemic countries to optimize program implementation [9]. Since STH and SCH NTD control programs rely on MDA, it is crucial to comprehend the factors that impact the different stages of implementation, such as acceptability, coverage, and compliance. Moreover, identifying and assessing the factors that influence implementation outcomes is essential to ensure the successful implementation of NTD control programs. In the past, Ogun State has implemented MDA programs separately for SCH and STH before introducing an integrated school-based MDA program for both diseases in 2016 [11]. Despite significant progress in eliminating SCH and STH, there is a need to investigate the implementation challenges that hinder the uptake, uptake, and coverage of the integrated MDA programs at the community level. Understanding the community's perceptions can provide valuable insights for program managers to improve current control strategies [12].

The main objective of this study was to explore the community's perceptions of the school-based MDA control programs for STH and SCH infections and to identify potential barriers to uptake and coverage.

## Methods and materials

### Ethical statement

The study was conducted following the principles outlined in the Declaration of Helsinki, and the study participants' data were treated with strict confidentiality throughout the study. The study received ethical approval from the Institutional Review Board of the Nigerian Institute of Medical Research, Lagos, Nigeria (reference number IRB/21/004, approval date February 16, 2021), and social approval from the Department of Health Planning, Research, and Statistics, Ogun State Ministry of Health, Ogun State, Nigeria (reference number HPRS/351/388, approval date June 16, 2021). Verbal informed consent was obtained from all study participants after the objective and purpose of the study were explained. Informed assent forms were provided to children and minors and informed parental consent was obtained for their parents or local guardians. For illiterate participants, impartial witnesses were enlisted to assist with the consenting process. Participants were included in the study only after individually signing the informed consent/assent form and receiving a paper copy.

### Study settings

Ogun State, Nigeria, was selected as the study location due to its high prevalence of STH (19.2%) and SCH (32.2%) and the inconsistent support of partner programs for NTD control. Four local government areas (LGAs) in Ogun State were purposively selected based on recommendations from the Department of Public Health, Ogun State Ministry of Health. The following local government areas (LGAs) were selected for the study: Ikenne, with a prevalence of 12.6% for STH and 1.46% for SCH; Abeokuta North, with a prevalence of 22.4% for STH and 15.40% for SCH; Abeokuta South, with a prevalence of 27.5% for STH and 1.61% for SCH; and Obafemi Owode, with a prevalence of 21.1% for STH and 8.78% for SCH [13].

## Study design and participants

This study employed a qualitative research approach to explore barriers to the uptake and coverage of control interventions in neglected tropical disease (NTD) control programs. The study focused on stakeholders involved in NTD control programs in Ogun State, Nigeria, where STH and SCH are endemic. Purposive sampling was used to select key informants and focus group participants based on their engagement in various positions in the health system and community and their involvement in the management of NTDs. The study conducted in-depth Key Informant Interviews (KIIs) with community members such as the NTD focal persons. Additionally, the study conducted Focus Group Discussions (FGDs) with parents, teachers, town announcers, opinion leaders, enrolled school-aged children (ESAC), and non-enrolled school-aged children (NESAC). A semi-structured questionnaire was used during these sessions to guide the exploration of ideas. Thematic analysis was used to extract key themes and sub-themes from the data. This methodology enabled a comprehensive understanding of the challenges faced by NTD control programs in addressing STH and SCH and provided valuable insights into the barriers to achieving effective control interventions.

## Sample size determination

As a qualitative study, the authors did not perform sample size calculation; instead, they predetermined the number of Focus Group Discussions (FGDs) and Key Informant Interviews (KIIs) necessary to achieve ideal saturation. The authors planned to conduct 12 FGDs and 15 in-depth interviews to reach this goal.

## Data collection

A total of 144 community participants were involved in the study. A total of 12 FGDs (ESAC, NESAC, community members, teachers, and parents) and 16 KIISs (Teachers (Head of school), Parents (some selected parents), and Community members (Head of the community, opinion leader, town announcers) were conducted amongst the community groups who have enrolled SAC and Non-enrolled SAC; teachers and parents; opinion leaders and town announcers. The semi-structured and FGDs interview topics were developed and pretested (See S1–S3 Files) to ensure the data's reliability. Audio tape recorders captured all the information obtained during the KIIs and FGDs. The interviewers also made observations and took notes while recording to ensure accuracy. The interviews were conducted in both English (the national language of Nigeria) and Yoruba (the local language of the study participants) in various settings, including small halls, private rooms, and offices. The duration of each KII ranged from 20 to 40 minutes. FGDs were conducted in the participants' respective LGAs.

## Data processing and analysis

Data from FGDs and KIIs were transcribed, coded, and analyzed thematically based on the emerging themes of barriers, drugs, knowledge, and recommendations. The recorded audio in the Yoruba language was transcribed into the English language. The team members checked the transcription independently for verification and accuracy with simultaneous audio playing. The study content was analyzed thematically, indexed, and coded inductively using the QRS Nvivo 12 software package (QRS International, Doncaster, Australia) for the content analysis of unstructured qualitative data. The initial open codes were sorted into sub-themes based on their similarity. According to the participant's responses, these sub-themes were clustered and refined to form broad themes and debated within the research team. Data collection, analysis, and reporting followed the Standards for Reporting Qualitative Research (SPQR) guidelines.

Guba and Lincoln's criteria for determining rigor in qualitative research were used to ensuring consistency during protocol preparation, data collection, development of a coding system, interrater reliability, and data analysis [14]. Two interviews per group were coded by two authors (FT, DN), for which the degree of similarity was determined by calculating the interrater reliability using the QRS Nvivo 12 software package (QRS International, Doncaster, Australia). Cohen proposed that Kappa scores should be interpreted using the following: values ≤ 0 indicate lack of agreement, values 0.01–0.20 indicate no or little agreement, values 0.21–0.40 indicate fair agreement, values 0.41–0.60 indicate moderate agreement, values 0.61–0.80 indicate substantial agreement, and values 0.81–1.00 indicate nearly perfect agreement. Kappa's score for the thematic analysis was 1.00, indicating nearly perfect agreement.

## Data quality control

The team members discussed the KIIs and FGDs questions before data collection to ensure the data quality. The study team organized three-day training for the data collectors on how to conduct in-depth interviews and FGDs. Supervision was done throughout the data collection process by the investigators.

## Results

The study findings were split into four thematic areas: Perception, Drug, Barriers, and Recommendations, as shown in Table 1. The participants were drawn from different interest groups whose socio-demographic characteristics demonstrated the appropriateness of each category. The study participants described key components of MDA-integrated control programs, such as the critical role of teachers, parents, and community members in engaging MDA in terms of mobilization and sensitization.

## Demographic characteristics of participants

A total of 144 community participants were involved in the study. A total of 12 FGDs were conducted amongst the community groups who have enrolled SAC and Non-enrolled SAC; teachers and parents, opinion leaders, and town announcers. Table 2 below presents the composition of the KII and FGD, and Table 3 the socio-demographic characteristics of the study participants in the KIIs and FGDs.

## Theme 1: Perception

Most of the community members interviewed had adequate knowledge of the symptoms of schistosomiasis and soil-transmitted helminths.

**Table 1. Themes and Sub-themes.**

| Themes | Sub-themes |
|---|---|
| **Perception** | 1. Mode of Transmission of STH and SCH<br>2. Awareness |
| **Drugs** | 1. Safety<br>2. Acceptability |
| **Barriers** | 1. Influence of Parents<br>2. Lack of Sufficient Knowledge<br>3. Adverse Drug Reactions |
| **Recommendation** | 1. Community Sensitization and Engagement<br>2. Drug Distribution and Performance<br>3. Partner Collaboration and Coordination |

**Table 2. Composition of Key Informant Interview and Focused Group Discussion.**

| Participants | Female | Male | Total |
|---|---|---|---|
| Enrolled and non-enrolled school-aged children | 31 | 17 | 48 |
| Teachers/Parents | 31 | 17 | 48 |
| Community members | 26 | 22 | 48 |

**Sub-theme 1.1: Mode of Transmission.** When asked about the mode of transmission of STH infection, an ESAC said that,

*"urinating in the river to prevent it; we should not be urinating again."*

Another NESAC said that,

*"Not good to pee around, especially where the dog has peed. It will lead to bloody urine."*

While another NESAC said that,

**Table 3. Socio-demographic characteristics of the study participants in the KIIs and FGDs.**

| Description | Frequency (N = 144) | Percentage (%) |
|---|---|---|
| **Gender** | | |
| Male | 56 | 39.0% |
| Female | 88 | 61.0% |
| **Age in years** | | |
| 6–9 | 8 | 5.60% |
| 10–14 | 30 | 20.8% |
| 15–19 | 10 | 6.90% |
| 20–24 | 9 | 6.30% |
| 25–29 | 8 | 5.60% |
| 30–34 | 25 | 17.40% |
| 35–39 | 20 | 13.9% |
| 40–44 | 18 | 12.5% |
| 45–49 | 12 | 8.30% |
| ≥ 50 | 4 | 2.80% |
| **Marital Status** | | |
| Single | 62 | 43.1% |
| Married | 77 | 53.5% |
| Divorced | 5 | 3.50% |
| **Religion** | | 0.0 |
| Christianity | 79 | 54.9% |
| Islam | 60 | 41.7% |
| Missing | 5 | 3.50% |
| **Occupation** | | 0.0 |
| ESAC | 24 | 16.7% |
| NESAC | 24 | 16.7% |
| Parents | 24 | 16.7% |
| School teachers | 24 | 16.7% |
| Opinion leaders | 24 | 16.7% |
| Town announcers | 24 | 16.7% |

*"STH infection can be contacted through water if an infected person has peed inside the river."*

The most common perception about the transmission of STH infection was,

*"Do not pee in the river to prevent."*

An ESAC said that

*"It is bad for someone to urinate everywhere because wherever a dog urinates, and a person goes to urinate there, such an individual can be infected by passing out bloody urine."*

Most of the N/ESAC opined that the disease is caused by contact with dog urine.

*"By urinating on the same spot, a dog urinates."*

*"By urinating in the same spot a dog had urinated and bathing inside an infected river."*

A NESAC participant stated that schistosomiasis is

*"A disease that worries the stomach if one hasn't eaten."*

Another NESAC opined that,

*"It worries the stomach if the drug is not used properly."*

While a few others attributed the causation to the consumption of sugary things,

*"If one eats many sugary things and eats dirty things."*

*"Eating sugary things in the morning."*

An ESAC said that

*"Worm causes malaria. It is a bad disease that happens in Nigeria."*

Another ESAC said that,

*"What I know about worm disease is that it does not allow our body to react as it should. It also caused vomiting and dry body."*

When asked if it could be transmitted to another person, most NESAC state that

*"It cannot be transferred from one person to another, especially by touch."*

Only one person opines that body fluid may transmit it from one person to another.

*"If the blood or saliva of an infected person touches the saliva or blood of an uninfected person, they can contract the disease."*

**Sub-theme 1.2: Awareness.**   Regarding the awareness of the drug given to them, most of the ESAC and NESAC spoke about the color of the drug, but only a few of them could mention the drug's name. Although, the teachers and some parents could provide the names of the drugs.

A NESAC student stated that,

*"The administrators ask about our ages and measure our heights, corresponding to the drug type we will be given. Generally, these tablets are of different shapes: one is slightly longer while the other is round."*

An ESAC stated that,

*"It's white in color and long, and the name of the drug is Albendazole."*

According to an opinion leader, different drugs are given for STH and SCH. He mentioned some of the drugs.

*"Mectizan for eye worms, Albendazole for worms, mebendazole for Schistosomiasis."*

Another opinion leader mentioned the excellent awareness of the MDA program.

*"It is known that once or twice a year, the MDA program will take place, so parents must ensure their children eat well because the drug is strong."*

When questioned about drug awareness, most teachers stated that the State Health Department would broadcast radio advertisements as the drug administration period approached. Additionally, stakeholders at the community level would be informed.

*"Except you do not listen to the radio alone; that is when you may miss out on the jingles."*

One parent noted that when the MDA program commences,

*"We see health workers, often clothed in identifiable uniforms and sometimes using megaphones, urge parents to bring their children and dependents to receive the medication."*

An opinion leader also expressed that the community acknowledges the government's efforts, which they consider effective to a certain extent, stating:

*"Yes, they are given through house-to-house administration and mobilization to health centers."*

Parents encouraged the government to increase public awareness of the drug and the program.

*"Government should continue to do more awareness to people."*

*"Awareness creation and educating them more, on media about the drug"*

*"Continue doing the publicity because most parents are ignorant of the problem."*

In contrast to the government-provided treatment, several respondents discussed the availability of alternative treatments within the community. One parent remarked,

*"People get treated through herbal treatment (Yoruba herbs) and clinic care."*

Another parent elaborated on the alternative treatment offered by traditional healers in their community. She asserted that while this treatment is effective, it presents a challenge regarding the dosage of herbs, which cannot be accurately measured in the same way as a clinical prescription. This perspective is captured in the following quote from a parent.

*"The herbs, our traditional healers use herbs for it. They cook and drink it without measurement, and as they are drinking it, it will just come out through urine or feces."*

Another parent also felt that,

*"I always use the modern method because we have been advised to do so. In addition, locally made drugs do not have dosage, which might even cause more harm to our children than good. Therefore, I believe modern medication is right for these diseases".*

An opinion leader said,

"*I do not go for herbal concoction. In the hospital, there is no*

significant cost, just card fees."

## Theme 2: Drug

**Sub-theme 2.1: Safety.**   When questions about the safety and efficacy of the drugs were asked, an ESAC stated that,

*"The drug is powerful; you must be strong before taking it."*

A popular scenario that occurred a few years ago was recounted by one of the ESAC, stating that

*"A female student slept after taking the drugs till closing time, and the next*

*the day we were told she was dead."*

The event's details were unknown, but it was linked to using the strong drug given in school. However, an opinion leader debunked the story during the FGD sessions.

*"Though there were rumors and rejection, people were later educated on the drugs, and it only administered a few hours after breakfast."*

It is important to note that some parents were against using the drug because of its strength; hence they discourage their children from taking it when given in school.

*"There are some side effects, but it has been effective. It is a potent drug, and it causes dizziness."*

Most parents are unsure about the drug's safety due to concerns about its potential side effects on their children.

**Sub-theme 2.2: Acceptability.**    The community broadly embraced the MDA program. An opinion leader said,

"*Monthly meetings are organized to foster socialization, and there's a strong collaboration between us regarding the MDA.*"

Another opinion leader confirmed that the community members cooperate with them to make the MDA program effective. He stated that,

"*Although there is no direct communication, we connect through representatives, and the cooperation is effective.*"

A town announcer said,

"*Through the records, it is of high acceptability, and there has been positive support from community members.*"

A teacher who was involved in the program shared her experience, showing improvement in the MDA programs.

"*The program is effective. I say this because I received over 400 doses of the drug last year, but only 100 were used due to rejection. However, this year, thankfully, I put in extra effort, explaining the benefits to the parents, and the acceptance increased. Hence, I believe the program is effective.*"

A parent affirmed that the medication was provided to them free of charge through the program.

"*The government provides medication free of charge for treating STH and schistosomiasis.*"

*2.2.1 Side Effects Experience*. The primary issue raised when asked about the acceptability was the side effects of the drugs, which motivates some mothers to discourage their children from taking the drugs, but a more significant portion of the parents said they have no issues accepting the drugs after considering the benefits they stand to gain from it. FGDs among the ESAC and NESAC revealed rumors, and parents' opinions substantially affect drug acceptability. Based on the responses from the NESAC, it was clear that many harbored skepticism about the drug and its potential side effects. The ESAC expressed similar concerns regarding the drug's side effects.

An ESAC recounted an adverse reaction his brother experienced after taking the medication.

"*Because it made my brother faint, which made him return home the next day, and her parent warned her not to use it again.*"

An ESAC stated in an FGD section that he accepted the drug because a friend accepted it, and there were no side effects.

"*Because she had used it before, and it didn't affect her.*"

However, some NESAC said they would not accept the drug stating some reasons which were mainly related to the side effects of the drugs. Some of the side effects highlighted were:

*"Fear of vomiting, bitter taste, and laziness."*

*"Some people think the drug might make them vomit or stool."*

*"Some set of people that received these medications complained of headache."*

*"Some people already on medications refused to take these drugs, and some felt taking them would weaken them."*

*"Yes, it is very effective, though it made some kids dizzy, it's very effective."*

A parent acknowledged there were side effects though the drug is effective.

*"Despite some side effects, the drug is effective and potent, leading to successful outcomes. However, taking it without consuming substantial food may induce dizziness or fainting. In such instances, individuals are advised to consume glucose."*

It was noted that a mother's decision significantly influenced the drug's acceptance. The following quotes from the N/ESAC illustrate this point,

*"Last year, she fell ill, so her mother said never to retake it."*

*"Because their mum asked them not to use it, that it will make them fall sick."*

Some opinion leaders suggested that obtaining the drug from a private healthcare facility might lend it more credibility. They also mentioned the financial ability of parents to purchase the drug outside the state government's MDA periods. One participant noted that some religious leaders advocate against the use of medication. These are crucial considerations for the MDA providers to address.

## Theme 3: Barriers

The study evaluated perceived barriers among participants regarding the acceptance of the drugs.

**Theme 3.1: Influence of parents.**    The ESAC pointed out that,

*"People decline because of parents' instructions because their mum asked them not to use it that it could make them sick."*

*"The fear of vomiting and laziness, some children throw the drug away because it is bitter."*

*"The smell of the drug, it is not nice."*

A NESAC opined that,

*"Some parents warned their children not to receive these medications, saying the medications are bad."*

*"The smell of drugs is not nice; I fear vomiting, bitter taste, and laziness."*

Interestingly, another NESAC recommended that the medication be more pleasant to take, enhancing its performance and encouraging parents to persuade their children to use it.

*"If the drug is sweet, if it works well in the body if parents insist not to use it, I will use it."*

**Theme 3.2: Inadequate knowledge.**   In the FGDs among the parents, one of the parents stated that,

*"Some parents are still ignorant about it; they do not know what the drug is about."*

Some recommended educating parents and even the kids on the benefit of taking the drug.

*"Inform the child and parents about the performance of the drug."*

*"When you educate and sensitize them and also give them gifts."*

**Theme 3.3: Demeanour and incentives.**   Others stated the side effects of the drugs as barriers to the acceptability of the drug. Interestingly, a parent emphasized an essential barrier affecting the drug's acceptability to her and some others.

*"Appearance shows the manner. Sometimes the medical staff appears moody to the parents and people. This will discourage the masses from accepting what is brought to them."*

As the gatekeepers of the communities, the viewpoints of opinion leaders and town announcers were sought. They identified several potential obstacles to the acceptance of the drug, insights that could aid providers in strategizing for its distribution. Transportation of the drug was underscored as one of the challenges encountered. Another potential issue was the delay in announcing or raising awareness about the MDA roll-out.

*"Time allowed for campaign and education is not enough."*

Another participant noted that incentives given to drug administrators are essential.

*"The small incentive is discouraging. When there was no payment, we didn't go around, but the turnout was poor. People said we didn't give them food to take medicine. But now we do house-to-house. The money is small, no doubt, but it will be encouraging*

*if paid on time."*

A participant noted that rumors about the drug were a barrier to the acceptability of the drug, stating that:

*"These things are a drawback to this program because some rumors are untrue and didn't happen."*

## Theme 4: Recommendation

Suggestions on how to improve the program's performance and impact were gathered from thought leaders and other participants in the study.

**Sub-theme 4.1: Community sensitization and engagement.**   NESAC mentioned the importance of financial incentives to people for using the drug.

*"Give kids incentives."*

Similarly, ESAC also suggested giving incentives.

*"Sweets and candies should be shared with the children while distributing the drugs to them; they should be cajoled into taking it by promising the nice things, e.g., biscuits, sweets and telling them the drug is sweet."*

*"By buying things for the children, biscuits sweets. . . even the reluctant children will accept the drug afterward."*

Some others, such as teachers and parents, opined that,

*"Awareness through radio programs and media for people to know the efficiency and performance of the drug."*

*"Providing education about the medication to ensure they understand its efficacy."*

**Subtheme 4.2: Dug Distribution and Performance.**  Feedback on enhancing the distribution of the drugs indicated that partnerships with private pharmacies could significantly expand the drug's reach. In addition, as noted earlier, a thought leader suggested involving private healthcare facilities could boost the program's credibility. Some individuals harbor biases against government health facilities and programs and might trust and engage more with a private sector partnership.

*"Incorporating private healthcare providers into the program could enhance its credibility."*

Another crucial aspect to consider is the logistics of drug distribution. A local town announcer pointed out that some remote areas are difficult to reach, which may cause health workers to skip these locations if they lack sufficient motivation. To address this issue, the announcer recommended providing motorcycles to help the health workers access these challenging areas.

*"The accessibility of some villages is hindered by poor roads, making them unreachable by car. Providing bicycles could access these areas, and the necessary work could be carried out."*

An opinion leader also noted that the current compensation for the workers might not be sufficient to motivate them to overcome challenging conditions while distributing the medications.
Some parents recommended some proactive measures.

*"Appearance shows the manner. Sometimes the medical staff appears moody to the parents and people. This will discourage the masses from accepting what is brought to them."*

*"The medical staff should try to show love and joy while passing information across and explain the benefits well so people can understand."*

Other parents recommended food during the drug distribution.

*"If they can extend the drug's time and continuity and ensure food availability during the drug administration."*

*"To ensure there's the availability of food because the drug is strong."*

Opinion leaders echoed the same sentiment, strongly advocating for more vigorous awareness campaigns.

*"Megaphone should be given to creating awareness."*

*"Proper awareness through media to encourage people."*

*"Education of the drug to people."*

*"They should make awareness, and the drug should be made available and brought to every school and house."*

Some opinion leaders opined that the welfare of the frontline workers is taken more seriously.

*"The health care workers should be compensated and paid well for easy work and effective awareness."*

*"It is to ensure the health workers are paid, and proper awareness should be made."*

*"I will advise they pay health workers and take all things reported with all seriousness."*

*"Mobilizers need money and transportation too. Everything needs to be funded. Even the megaphones we use need a battery so that you can see money is key."*

*"Government should not relent and ensure funds are paid and do more awareness creation."*

Some opinion leaders suggested that MDA staff take the medication before the N/ESAC to promote its use.

*"The drug administrators should use the medicines first."*

Another opinion leader recommended that the exercise be repeated for students who missed the day of administration.

*"Mectizan should be twice a year because some are upset when they miss the drug distribution due to travel."*

*"The interval is too long; adequate education is required before administration."*

**Sub-theme 4.3: Partner collaboration and coordination.**   Community members stressed that if partnerships with private entities are established, the purchase of medications can extend beyond the set distribution period.

They also suggested involving religious leaders due to their significant influence over people. It was mentioned that some religious leaders advise against drug use, a critical issue for MDA providers to address.

To improve coordination, a town announcer highlighted the importance of involving field officers in the initial planning stages, emphasizing that their input would be invaluable for smooth operation.

*"The Ministry of Health is refining its strategy, but involving the field officers more in this process is crucial. Since these officers have firsthand experience with on-the-ground challenges,*

*their insights are invaluable. Listening to their feedback and addressing their concerns could greatly improve organization and planning."*

## Discussion

Mass Drug Administration for vulnerable populations has been the best approach to prevent STH and SCH globally [15,16].

It was essential to find out if the children, the recipients of the drugs, were aware of the knowledge of the drugs and the process of MDA. It was difficult for the ESAC and NESAC to differentiate between STH and SCH infection and their transmission mode. The inadequacy of knowledge among the SAC has been substantiated by evidence from several African studies, for instance, studies done in Malawi, Kenya, Uganda, and Tanzania [17–21]. Poor environmental and sanitary conditions are linked to SCH infections [22], while STH is transmitted through contaminated soil [23,24]. Our findings revealed that most participants associated these diseases with drinking dirty, contaminated water. This finding is similar to the study conducted in Kenya, and the participants believed contracting schistosomiasis was associated with drinking dirty water and eating uncooked or contaminated food [18]." In addition, there were several misconceptions, like transmission through urinating, where dogs have urinated and bathed inside infected rivers. These findings agree with an Ethiopian study on low awareness and common misconceptions about Schistosomiasis [25]. Limited knowledge has been highlighted as a factor promoting the transmission of STH and SCH infection [26].

The majority of the participants demonstrated adequate knowledge of the symptoms of STH and SCH infection. They pointed out bloody stools and urine, vomiting, fever, headaches, abdominal aches, and generalized body aches.

It can be assumed that with a good knowledge of the drugs and the potential benefits, the acceptance would be better, and the attitude towards it would improve. Regarding the treatment of SCH and STH, most respondents were aware of the importance of the treatment and affirmed that the government provides drugs for them. The provision of drugs by the government is an advantage to these communities, as there is literature contrasting this with reports of drug shortages as challenges being encountered [27,28]. The N/ESAC recalled the color of the drugs in contrast to the adult respondents, who could state the difference in the drugs administered for STH and SCH. This could be because the N/ESAC are the primary beneficiaries of the school-based MDA control program.

STH and SCH were acknowledged as burdens among the community members, especially due to the knowledge deficit mentioned in several studies on the knowledge, attitudes, and practices of SCH and STH [18,19,27,29–31].

It was evident that people were involved in health programs in one way or another. The study revealed that the people in the community understand the implication of Schistosomiasis and soil-transmitted diseases, which makes them cooperate more and make the MDA program effective, thus underscoring the impact of MDA in the control of SCH and STH [24,32].

The acceptability of the administered drug during the MDA program was assessed among the ESAC and NESAC by asking whether the children would accept the drugs. It was noted that most of the NESAC accepted the drugs because of the benefit of the drug. Others did not accept the drug based on the color, bitterness, or sour taste; others mentioned side effects like headaches. These findings align with Kimani et.al. on the acceptability of praziquantel in treating *Schistosoma haematobium* in preschool children [33]. In addition, the family has a significant influence on the acceptability of MDA [34]. Our findings revealed that the influence of parents is significant to decision to accept the MDA. On the accessibility, the children could not tell categorically how it could be accessed; however, the parents and the teachers expressed

that the drugs were only administered by the health officers who summoned the parents to a meeting in school before the MDA exercise.

The parents and teachers voiced that fears about the drugs and acceptability of the drug hinder the acceptance and uptake of the MDA program among them, even as stakeholders to the children. In addition, most parents and teachers noted a need for further enlightenment on the program and the effects of the drug especially. Further, our findings revealed drug side effects, parental instructions against usage, and unpalatable drug characteristics as barriers. Other studies have identified similar barriers [27, 32, 35].

The study participants fronted some recommendations, categorized as improving the welfare package of the frontline health workers on the program, revisiting the distribution strategy, increasing awareness levels, and giving incentives to children. In addition, there is a need for clear, concise, and consistent sensitization to the communities [18, 29, 32].

In this community perspective study, we found that participants generally had correct perceptions of STH and SCH transmission. However, some misperceptions persisted regarding disease causation and transmission, as well as fear and distrust surrounding the deworming drugs used in mass drug administration (MDA). Specifically, participants were concerned about handling side and adverse effects, the capabilities of MDA providers to address them, and perceived inferiority to commercially available anthelmintic drugs. Despite these challenges, participants acknowledged a knowledge gap and desired better sensitization campaigns, information dissemination, and safety assurances. To address these issues, we recommend strengthening health education messages and increasing the visibility or availability of on-site medical personnel. By understanding how parents and children perceive MDA activities, policymakers and program implementors can work to eliminate barriers to compliance and achieve the target coverage rate for administering chemo-preventive drugs for intestinal parasites. Overall, our findings have important implications for improving MDA programs and increasing their performance in reducing the burden of intestinal parasitic infections in this community.

The study had some limitations. Firstly, it was conducted in Ogun State, Nigeria communities; therefore, the findings may not be generalizable to other populations or communities with different socioeconomic or cultural backgrounds. Secondly, participants may have provided socially desirable responses, leading to potential social desirability bias. Thirdly, the study relied on self-reported data, which may have been subject to recall bias or misreporting. Fourthly, the study only collected cross-sectional data at a single point in time, and longitudinal data would have allowed for a better understanding of changes in community perceptions and behaviors over time.

## Supporting information

**S1 File. Key Informant Interview Guide.**
(PDF)

**S2 File. Focus Group Discussion Guide for Parents, Teachers, Opinion Leaders, and Town Announcers.**
(PDF)

**S3 File. Focus Group Discussion Interview Guide for School-aged Children.**
(PDF)

## Author Contributions

**Conceptualization:** Folahanmi T. Akinsolu, Olaoluwa P. Akinwale.

**Data curation:** Folahanmi T. Akinsolu, Olunike R. Abodunrin, Mobolaji T. Olagunju, Diana W. Njuguna, Abideen O. Salako.

**Formal analysis:** Folahanmi T. Akinsolu, Mobolaji T. Olagunju.

**Investigation:** Folahanmi T. Akinsolu, Mobolaji T. Olagunju, Nurudeen O. Rahman, Islamiat Y. Soneye, Abideen O. Salako, Oliver C. Ezechi, Orsolya E. Varga, Olaoluwa P. Akinwale.

**Methodology:** Folahanmi T. Akinsolu, Olunike R. Abodunrin, Mobolaji T. Olagunju, Nurudeen O. Rahman, Diana W. Njuguna, Islamiat Y. Soneye, Abideen O. Salako, Orsolya E. Varga, Olaoluwa P. Akinwale.

**Project administration:** Mobolaji T. Olagunju, Islamiat Y. Soneye, Abideen O. Salako, Olaoluwa P. Akinwale.

**Resources:** Nurudeen O. Rahman.

**Software:** Mobolaji T. Olagunju, Ifeoluwa E. Adewole, Anita M. Dabar, Diana W. Njuguna.

**Supervision:** Folahanmi T. Akinsolu, Oliver C. Ezechi, Olaoluwa P. Akinwale.

**Validation:** Folahanmi T. Akinsolu, Mobolaji T. Olagunju, Ifeoluwa E. Adewole, Nurudeen O. Rahman, Anita M. Dabar.

**Visualization:** Diana W. Njuguna, Oliver C. Ezechi.

**Writing – original draft:** Folahanmi T. Akinsolu, Olunike R. Abodunrin, Mobolaji T. Olagunju, Ifeoluwa E. Adewole, Diana W. Njuguna, Islamiat Y. Soneye, Abideen O. Salako, Oliver C. Ezechi, Orsolya E. Varga, Olaoluwa P. Akinwale.

**Writing – review & editing:** Folahanmi T. Akinsolu, Olunike R. Abodunrin, Mobolaji T. Olagunju, Anita M. Dabar, Diana W. Njuguna, Islamiat Y. Soneye, Abideen O. Salako, Oliver C. Ezechi, Orsolya E. Varga, Olaoluwa P. Akinwale.

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
