## [Decision Letter · Decision Letter 0]

25 Apr 2023

Dear Dr. Akinsolu,

Thank you very much for submitting your manuscript "Community Assessment of School-Based Mass Drug Administration Program for Soil-Transmitted Helminths and Schistosomiasis in Nigeria." for consideration at PLOS Neglected Tropical Diseases. As with all papers reviewed by the journal, your manuscript was reviewed by members of the editorial board and by several independent reviewers. In light of the reviews (below this email), we would like to invite the resubmission of a significantly-revised version that takes into account the reviewers' comments. 

We cannot make any decision about publication until we have seen the revised manuscript and your response to the reviewers' comments. Your revised manuscript is also likely to be sent to reviewers for further evaluation.

Sincerely,

Uwem Friday Ekpo, PhD

Academic Editor

Eva Clark

Section Editor

Reviewer's Responses to Questions

**Key Review Criteria Required for Acceptance?**

**Methods**

-Are the objectives of the study clearly articulated with a clear testable hypothesis stated?

-Is the study design appropriate to address the stated objectives?

-Is the population clearly described and appropriate for the hypothesis being tested?

-Is the sample size sufficient to ensure adequate power to address the hypothesis being tested?

-Were correct statistical analysis used to support conclusions?

-Are there concerns about ethical or regulatory requirements being met?

Reviewer #1: these are attached

Reviewer #2: The objective was clearly stated and study appropriate to address the stated objective. The study population was clearly described. The study has a sufficient sample size. Analysis done was adequate. No concerns.

**Results**

-Does the analysis presented match the analysis plan?

-Are the results clearly and completely presented?

-Are the figures (Tables, Images) of sufficient quality for clarity?

Reviewer #1: these are attached

Reviewer #2: It does. The results presented are clear. The tables and figures are okay.

**Conclusions**

-Are the conclusions supported by the data presented?

-Are the limitations of analysis clearly described?

-Do the authors discuss how these data can be helpful to advance our understanding of the topic under study?

-Is public health relevance addressed?

Reviewer #1: these are attached

Reviewer #2: The conclusions are supported by the data and findings. The limitations were clearly stated and described. The public health relevance was addressed.

**Editorial and Data Presentation Modifications?**

Reviewer #1: these are attached

Reviewer #2: Minor revisions are recommended, hence accept after the authors address the issues raised.

**Summary and General Comments**

Reviewer #1: these are attached.

Reviewer #2: The manuscript was well articulated and presented. The authors may further explore the stakeholders perception about availability of drugs during MDA, timeliness in drug supply, distribution and or inequity in distribution.

PLOS authors have the option to publish the peer review history of their article (what does this mean?). If published, this will include your full peer review and any attached files.

Reviewer #1: No

Reviewer #2: Yes: Dr Obiageli Josephine Nebe
---

## [Decision Letter · Decision Letter 1]

15 Jun 2023

Dear Dr. Akinsolu,

We are pleased to inform you that your manuscript 'Community Perception of School-Based Mass Drug Administration Program for Soil-Transmitted Helminths and Schistosomiasis in Ogun State, Nigeria.' has been provisionally accepted for publication in PLOS Neglected Tropical Diseases.

Best regards,

Uwem Friday Ekpo, PhD

Academic Editor

Eva Clark

Section Editor

Reviewer's Responses to Questions

**Key Review Criteria Required for Acceptance?**

**Methods**

-Are the objectives of the study clearly articulated with a clear testable hypothesis stated?

-Is the study design appropriate to address the stated objectives?

-Is the population clearly described and appropriate for the hypothesis being tested?

-Is the sample size sufficient to ensure adequate power to address the hypothesis being tested?

-Were correct statistical analysis used to support conclusions?

-Are there concerns about ethical or regulatory requirements being met?

Reviewer #2: The objectives are well articulated with clear testable hypothesis. The study design is quite appropriate and addressed the objectives. Well described study population. The sample size is fine for a quantitative study of this nature. It was more of descriptive analysis as the study is purely a quantitative one. No ethical concerns.

**Results**

-Does the analysis presented match the analysis plan?

-Are the results clearly and completely presented?

-Are the figures (Tables, Images) of sufficient quality for clarity?

Reviewer #2: The descriptive analysis matched the analysis plan. The results are clearly presented. The tables are of sufficient quality.

**Conclusions**

-Are the conclusions supported by the data presented?

-Are the limitations of analysis clearly described?

-Do the authors discuss how these data can be helpful to advance our understanding of the topic under study?

-Is public health relevance addressed?

Reviewer #2: The conclusions are supported by the tables presented. The limitations are clearly described. The authors reflected that in the manuscript. It was addressed.

**Editorial and Data Presentation Modifications?**

Reviewer #2: Accept after minor corrections.

**Summary and General Comments**

Reviewer #2: The manuscript is well written in an intelligible and scholarly English. It should be accepted for publication after minor revision.

PLOS authors have the option to publish the peer review history of their article (what does this mean?). If published, this will include your full peer review and any attached files.

Reviewer #2: **Yes: **Dr. Obiageli Josephine Nebe

---

## [Editor Report · Acceptance letter]

12 Jul 2023

Dear Dr. Akinsolu,

We are delighted to inform you that your manuscript, "Community Perception of School-Based Mass Drug Administration Program for Soil-Transmitted Helminths and Schistosomiasis in Ogun State, Nigeria.," has been formally accepted for publication in PLOS Neglected Tropical Diseases.

Best regards,

Shaden Kamhawi

co-Editor-in-Chief

Paul Brindley

co-Editor-in-Chief
